# Different gut microbial communities correlate with efficacy of albendazole-ivermectin against soil-transmitted helminthiases

Pierre H. H. Schneeberger [1,2 ✉], Morgan Gueuning[3], Sophie Welsche[1,2], Eveline Hürlimann[1,2], Julian Dommann[1,2], Cécile Häberli[1,2], Jürg E. Frey[3], Somphou Sayasone[2,4] & Jennifer Keiser [1,2 ✉]

Soil-transmitted helminth infections represent a large burden with over a quarter of the world's population at risk. Low cure rates are observed with standard of care (albendazole); therefore, a more effective combination therapy (albendazole and ivermectin) is being investigated but showed variable treatment efficacies without evidence of intrinsic parasite resistance. Here, we analyzed the microbiome of *Trichuris trichiura* and hookworm-infected patients and found an association of different enterotypes with treatment efficacy. 80 *T. trichiura*-infected patients with hookworm co-infections from Pak-Khan, Laos, received either albendazole ($n = 41$) or albendazole and ivermectin combination therapy ($n = 39$). Pre-/post-treatment stool samples were collected to monitor treatment efficacy and microbial communities were profiled using 16S rRNA gene sequencing, qPCR, and shotgun sequencing. We identified three bacterial enterotypes and show that pre-treatment enterotype is associated with efficacy of the combination treatment for both *T. trichiura* ($CR_{ET1} = 5.8\%$; $CR_{ET2} = 16.6\%$; $CR_{ET3} = 68.8\%$) and hookworm ($CR_{ET1} = 31.3\%$; $CR_{ET2} = 16.6\%$; $CR_{ET3} = 78.6\%$). This study shows that pre-treatment enterotype enables predicting treatment outcome of combination therapy for *T. trichiura* and hookworm infections.
Trial registration: ClinicalTrials.gov, NCT03527732. Registered 17 May 2018, https://clinicaltrials.gov/ct2/show/NCT03527732.

---

[1] Swiss Tropical and Public Health Institute, Department of Medical Parasitology and Infection Biology, Helminth Drug Development Unit, Basel, Switzerland. [2] University of Basel, Basel, Switzerland. [3] Agroscope, Research Group Molecular Diagnostics, Genomics and Bioinformatics, Wädenswil, Switzerland. [4] Lao Tropical and Public Health Institute, Vientiane, Lao People's Democratic Republic. ✉email: pierre.schneeberger@swisstph.ch; jennifer.keiser@swisstph.ch

Over 25% of the world's population is at risk of being infected by soil-transmitted helminths (STH)[1]. *Ascaris lumbricoides*, *Trichuris trichiura*, and hookworms (*Ancylostoma duodenale* and *Necator americanus*) account for most STH infections and the current recommended treatment is mainly based on two benzimidazole drugs, albendazole, and mebendazole, which display varying species-specific treatment efficacies[2]. For instance, a single-dose regimen of either drug will achieve poor cure rates treating *T. trichiura* infections[3,4] while displaying high efficacy against *A. lumbricoides* infections[5]. Moreover, mass drug administration (MDA) campaigns, which represent the main control strategy[6], can lead to increased selective pressure toward resistant parasites. Yet, the drug pipeline for new drug candidates remains empty despite recent efforts to discover novel drugs[7] and relies mainly on repurposing existing drugs from veterinary medicine[8]. The use of drug combinations, e.g., albendazole with ivermectin, has recently been described to broaden the spectrum of activity of treatments against soil-transmitted helminthiases[9]. Ivermectin is a derivate of avermectin, which was initially isolated in *Streptomyces avermectinius* and is commonly used in human and veterinary medicine[10]. It possesses a 16-membered macrocyclic lactone group and thus belongs to the macrolide drug class, which also encompasses antibiotic drugs such as azithromycin and erythromycin, used to treat mostly gram-positive bacterial infections. Ivermectin has been shown to improve the efficacy of single-dose treatment against *T. trichiura* when administered in combination with albendazole[11]. However, in a recent multi-country randomized controlled trial, varying treatment efficacy of albendazole-ivermectin against *T. trichiura* was observed with cure rates ranging from 13.8 to 65.7% in different settings[12].

One potential confounder of drug treatment efficacy is the dense and diverse non-parasitic gut microbiome[13]. Its pivotal role in treatment outcome is currently being investigated in different fields, including cancer research and metabolic diseases[14,15]. However, there is still very limited information pertaining to its role in the context of neglected tropical diseases and more specifically how it affects the treatment efficacy of essential drugs such as benzimidazoles for STH infections[16,17]. Various mechanisms of drug-microbe interactions have been identified but can broadly be categorized into either direct or indirect mechanisms[13]. The direct metabolization of drugs by gut microbes most often result in modulation of potency and/or toxicity and is a widespread mechanism[18]. Indirect mechanisms include, but are not limited to, local microbe-driven mechanisms of tampered gut wall function, resulting in deregulated drug translocation into systemic circulation[19,20].

In this study, we aimed to identify potential inter-kingdom mechanisms of resilience (i.e., bacteria modulating anti-parasitic drug efficacy) in the treatment of *T. trichiura* and hookworm infections. Understanding possible interactions between gut microbes and anthelmintic drugs could help to improve and optimize current treatment efficacy while avoiding selection towards resistant microbial communities. In the framework of this randomized trial, we assessed the efficacy of two anthelminthic treatments, albendazole alone (400 mg), and in combination with ivermectin (200 μg/kg), in the context of gut microbial communities. We conducted taxonomic profiling of gut microbial communities on pre-treatment stool samples to predict treatment efficacy assessed by post-treatment egg reduction rates. Microbial community composition was determined using several techniques, including 16S rRNA gene sequencing, 16S- and taxon-specific quantitative polymerase chain reaction (qPCR), as well as shotgun sequencing to identify species-level taxonomic features associated with treatment outcome.

## Results

**Sampling results and study characteristics.** Between January and May 2019, people from ten villages in Nambak, Luang Prabang district in Laos were tested for *T. trichiura*. 549 people were included and randomly assigned to either monotherapy (albendazole) or combination therapy (albendazole-ivermectin)[21]. Daily post-treatment stool samples were collected from a subset of 88 patients up to 28 days after first administration, among which 80 were included for microbiome assessment (Fig. 1). Among these, 86.2% (n = 69) were also co-infected with hookworms, 40% (n = 32) with *A. lumbricoides*, and 8.7% (N = 7) with *Opisthorchis viverrini*. Treatment arms were balanced in terms of sex, age, baseline infection intensities for any helminth species, and no noteworthy between-group differences were observed (Table 1). Similarly, sampling compliance in the post-treatment period was comparable between both treatment arms (Supplementary Fig. 1).

**Compositional clustering and cluster-specific taxonomic features.** Unsupervised clustering based on taxonomic composition before treatment enabled identification of underlying community structures into three distinct clusters, or enterotypes (ET1–3), as shown in Fig. 2A. The proportion represented by each enterotype was comparable between both treatment arms (Supplementary Fig. 2). The accuracy of this enterotype-based classification was 78.75% with a Kappa statistic of 62.93% when assessed using a random forest model on this dataset. Most important taxonomic features in terms of classification accuracy included the genera *Faecalibacterium* (2.704%), *Escherichia/Shigella* complex (2.613%), *Prevotella* (1.814%), *Phascolarctobacterium* (1.082%), and [*Eubacterium*] *coprostanoligenes* group (1.044%) (Fig. 2B). Relative abundances of *Faecalibacterium* and *Prevotella* were significantly higher in ET1 (P = 2.61E−09 and 7.55E−05, respectively), *Escherichia/Shigella* in ET2 (P = 1.33E−07), and [*Eubacterium*] *coprostanoligenes* group was enriched in ET3 (P = 7.55E−05) among a set of 8 genera found to be differentially abundant between the 3 enterotypes (Fig. 2C). The taxonomic composition was homogeneous when comparing samples from both treatment arms with a PERMANOVA ($r^2 = 0.013$,

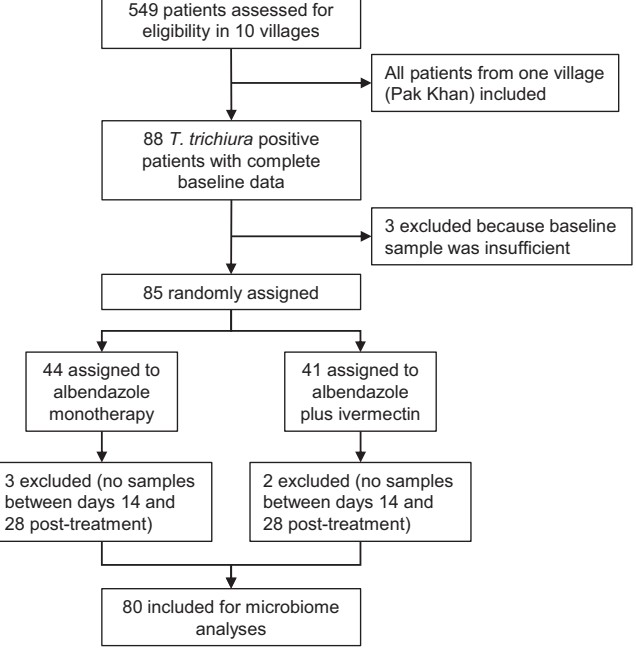

**Fig. 1 Trial profile.** Trial profile depicting the number of patients included for microbiome analyses.

$P = 0.38$), as shown in Fig. 2D. Comparison of alpha diversity showed significantly lower Shannon diversity and higher Berger–Parker dominance indices when comparing ET2 to ET1 and ET3 (Supplementary Fig. 3) but no differences were observed when comparing ET1 to ET3.

Taxon-specific absolute abundances and total bacteria confirm the presence of different community structures. Absolute abundance of *Faecalibacterium* (FAEC), *Escherichia* (ESCH), and *Prevotella* (PREV) assessed with qPCR confirmed the 16S rRNA gene sequencing-based enterotype classification (Fig. 3A). For instance, ESCH was significantly higher in ET2 than in both ET1 and ET3 ($P = 0.001$ and $0.002$). Similarly, FAEC and PREV were both significantly higher ($P = 0.001$ and $P = 0.004$, respectively) in ET1 than in ET3. We then tested whether qPCR values obtained from these three targets and total bacterial load (16S) could be used to classify samples into one of the three enterotypes identified by 16S rRNA gene sequencing. The qPCR-based random forest model accurately classified 83.1% of samples (Kappa = 0.72) into enterotypes 1–3, which is similar to the metrics obtained with the sequencing-based classification. The area under the receiver operating curve (AUC) of this model were of 1 (95% CI: N.A) between ET1 and ET2, 0.88 (95% CI: 0.79–0.97) between ET1 and ET3, and 0.91 (95% CI: 0.82–1) between ET2 and ET3 (Fig. 3B) indicating the suitability of qPCR-based classification into an identified enterotype as an alternative to high-throughput sequencing.

**Association between identified enterotypes and *T. trichiura* and hookworm cure.** We did not observe any association between cure rate, defined as an average egg per gram count (EPG) equal to zero between days 14 and 28 post-treatment, and ET, for patients receiving albendazole monotherapy for *T. trichiura* (Fig. 4A, right half) nor hookworm infections (Fig. 4B, right half). However, the majority of patients who received the combination therapy (albendazole-ivermectin) and were cured for both types of STH infections, presented a gut microbial composition classified as ET3 pre-treatment. This observation was similar for both, *T. trichiura* (Fig. 4A, left half) and hookworm infections (Fig. 4B, left half). Likelihood of cure of *T. trichiura* infection was significantly lower ($P = 0.0002$) for patients presenting ET1 than those presenting ET3 at baseline with an odds ratio (OR) of 0.03 (95% CI: 0.01–0.2). A similar, albeit weaker association ($P = 0.025$), with an OR of 0.14 (95% CI: 0.03–0.66), was observed between ET1/ET3 and treatment outcome of hookworm. We performed a similar analysis with a 90% egg reduction rate (ERR) as an estimate of treatment outcome. We found significant associations between enterotypes and ERR for both types of infection in the combination therapy arm similar to those found between enterotypes and cure rate (Supplementary Fig. 4). We did not find any associations between enterotype and ERR for patients receiving the monotherapy. Absolute abundances of PREV and FAEC measured at baseline correlated with average egg counts between days 14 and 28 post-treatment for the combination therapy with Spearman correlation coefficients of 0.416 ($P = 0.009$) and 0.275 ($P = 0.09$), respectively (Supplementary Fig. 5A). No correlation was observed between *T. trichiura* cure and absolute microbial abundances in the monotherapy arm, nor for hookworm in both treatment arms (Supplementary Fig. 5B–D).

**Species-level features enriched in either failure- or success-associated enterotype in the albendazole-ivermectin treatment arm.** To identify species-level differences between failure-associated enterotypes 1–2 and success-associated enterotype 3, we generated shotgun sequencing data from samples collected within the albendazole plus ivermectin treatment arm. In addition to previously observed enriched features from *Prevotella* (*P. copri* in ET1), *Faecalibacterium* (*F. prausnitzii* in ET1), *Escherichia/Shigella* (*E. coli*, in ET2), and *Ruminococcus* (*R. torques*, in ET3), we found *Streptococcus salivarius*, *Coprococcus eutactus*, *Roseburia faecis*, *Dorea longicatena*, and *Eubacterium halii* to be significantly enriched in either ET1 or ET3 when comparing both groups (Fig. 4C). *Coprococcus eutactus* was also depleted in ET3 when comparing it to ET2 while *Anaerostipes hadrus* was enriched in the former.

**Cox proportional hazard models of treatment outcome stratified by pre-treatment enterotype.** With two consecutive samples within the 28-days post-treatment period without eggs detected as the definition of cure of *T. trichiura*, measured hazard ratios indicate faster egg clearance while presenting ET3 (HR = 18.12, $P = 0.005$), when compared to ET1, for patients receiving the combination therapy. This association remains true when adjusting for potential confounders such as sex, age, and infection intensities (Table 2). Faster clearance is also observed for hookworm cure for ET3 (HR = 3.75, $P = 0.016$) when compared to egg clearance of patients presenting ET1. This association remains true when adjusting for sex, ($P = 0.04$), age ($P = 0.003$), infection intensity ($P = 0.017$), and baseline EPG ($P = 0.039$). The probability of early egg clearance was not different between patients presenting ET2 and ET1 for both, *T. trichiura* and hookworm infections, in the combination therapy arm. We did not observe any association between enterotype and probability of

**Table 1 Cohort description.**

| Characteristic | Albendazole-ivermectin | Albendazole | *P*-value |
|---|---|---|---|
| Number of patients (*n*) | 39 | 41 | |
| Age (y) | | | |
| Mean (arit) (±SD) | 29 (±17) | 26 (±17) | 0.37 |
| Range | 6-56 | 6-60 | |
| Female, *n* (%) | 25 (64.1%) | 20 (48.8%) | 0.18 |
| ATX (<1 month) | 0 (0%) | 2 (4.8%) | 0.49 |
| *Trichuris trichiura* infection | | | |
| Infected, *n* (%) | 39 (100%) | 41 (100%) | |
| EPG$_{arit}$ (±SD) | 1001 (±1427) | 793 (±884) | 0.45 |
| EPG$_{geom}$ (±SD) | 497 (±3.1) | 501 (±2.6) | |
| Range (EPG) | 102-6054 | 114-5004 | |
| I.i: Light, *n* (%) | 28 (71.8%) | 30 (73.1%) | |
| I.i: Moderate, *n* (%) | 11 (28.2%) | 11 (26.8%) | |
| *Hookworm infection* | | | |
| Infected, *n* (%) | 35 (89.7%) | 34 (82.9%) | |
| EPG$_{arit}$ (±SD) | 1418 (±1777) | 1548 (±3861) | 0.86 |
| EPG$_{geom}$ (±SD) | 766 (±3.2) | 358 (±5.9) | |
| Range (EPG) | 54-8574 | 12-22416 | |
| I.i: Light, *n* (%) | 28 (80%) | 27 (79.4%) | |
| I.i: Moderate, *n* (%) | 4 (11.4%) | 4 (11.8%) | |
| I.i: Heavy, *n* (%) | 3 (8.6%) | 3 (8.8%) | |
| *Ascaris lumbricoides* infection | | | |
| Infected, *n* (%) | 15 (38.5%) | 17 (41.5%) | |
| EPG$_{arit}$ (±SD) | 7992 (±8481) | 6739 (±9961) | 0.72 |
| EPG$_{geom}$ (±SD) | 2558 (±9.3) | 1423 (±10.2) | |
| Range (EPG) | 6-26,256 | 12-38,922 | |
| I.i: Light, *n* (%) | 8 (50%) | 12 (70.5%) | |
| I.i: Moderate, *n* (%) | 8 (50%) | 5 (29.5%) | |
| *Opisthorchis viverrini* infection | | | |
| Infected, *n* (%) | 2 (5.1%) | 5 (12.2%) | |
| EPG$_{arit}$ (±SD) | 15 (±3) | 39 (±35) | 0.45 |
| EPG$_{geom}$ (±SD) | 14.6 (±1.3) | 26.6 (±2.8) | |
| Range (EPG) | 12-18 | 6-108 | |

*y* year, *ATX* antibiotic treatment within one month before treatment, *EPG* eggs per gram of stool, *SD* standard deviation, *arit* arithmetic mean, *geom* geometric mean, *I.i* Infection intensity based on World Health Organization criteria.

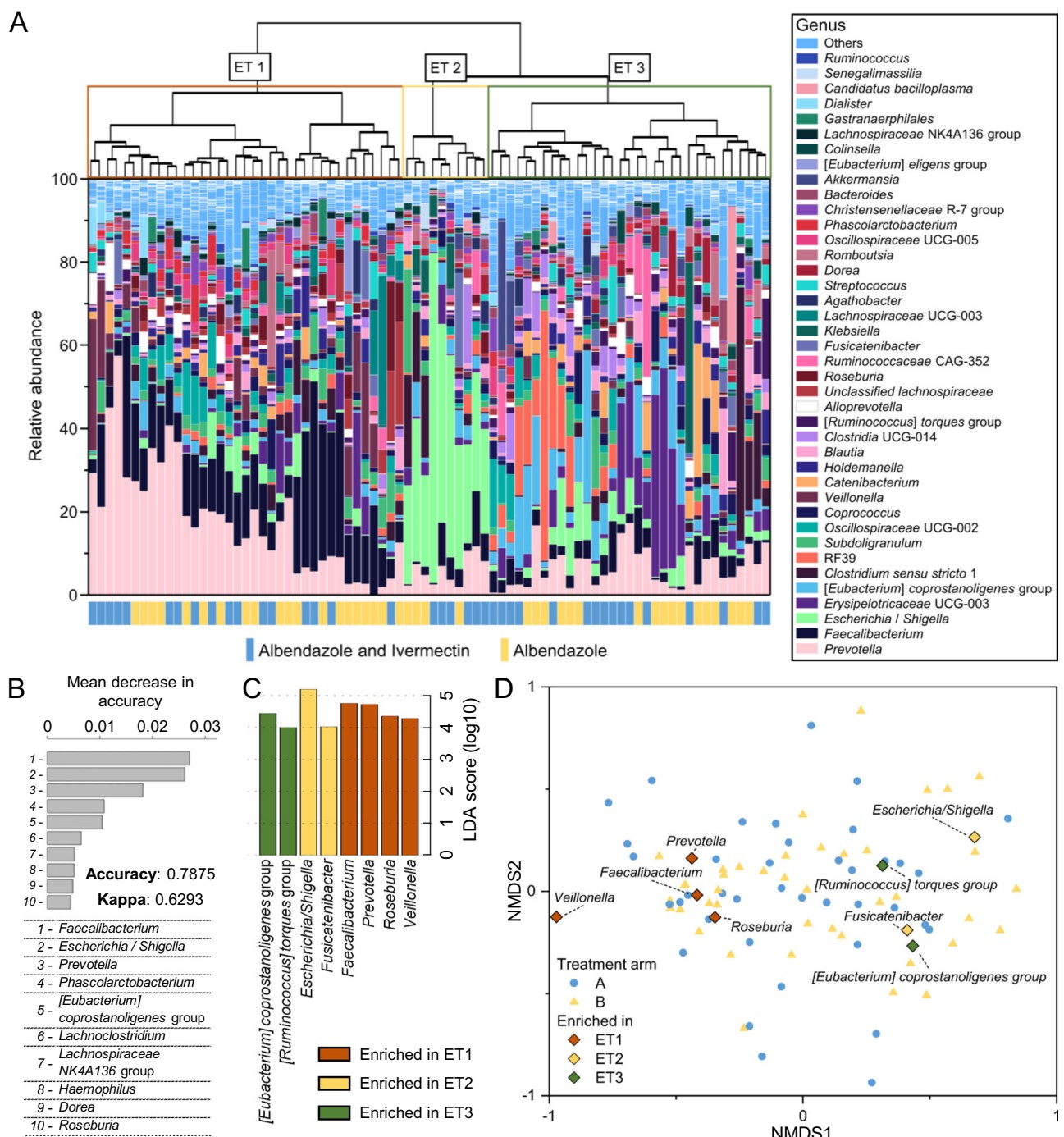

**Fig. 2 Underlying compositional structures and their taxonomic features. A** Gut microbial community composition of patients infected with *Trichuris trichiura* before treatment. The cladogram was generated using Bray–Curtis dissimilarity. **B** Performance of classification in this dataset using a random forest model. Taxonomic features are ranked according to their individual contribution to sample classification. **C** Bacterial genera found to be enriched in one of the enterotype using a Kruskal–Wallis test for group comparison combined with a linear discriminant analysis for effect size (Lefse). **D** Non-metric multidimensional scaling ordination plot of baseline samples using Bray–Curtis dissimilarity. The labeled genera were found to be enriched in either enterotype. Treatment arm A = albendazole and ivermectin, treatment arm B = albendazole, ET enterotype.

being cured in the monotherapy arm. Pre-/post-treatment EPG counts strongly correlated in the monotherapy arm but did not correlate in the combination therapy arm, which is consistent with the observed treatment efficacies (Supplementary Fig. 6).

Associations between success- and failure-associated enterotypes and daily post-treatment eggs per gram of stool counts. Using a survival analysis, we confirm that patients presenting ET3 pre-treatment are more likely to be both, faster and more

efficiently cured of a *T. trichiura* infestation using the albendazole and ivermectin-based treatment than those presenting ET1 and ET2 (*P* = 0.0002; Fig. 5A, left-half). The same observation applies in the context of hookworm infections (*P* = 0.009; Fig. 5A, right-half). Overall efficacy of 33.3% against *T. trichuris* was observed. ET-specific efficacy was 5.8% for patients with ET1, 16.6% for patients with ET2, and 68.8% for patients with ET3. Overall efficacy against hookworm was 47.2%. Patients with ET3 revealed

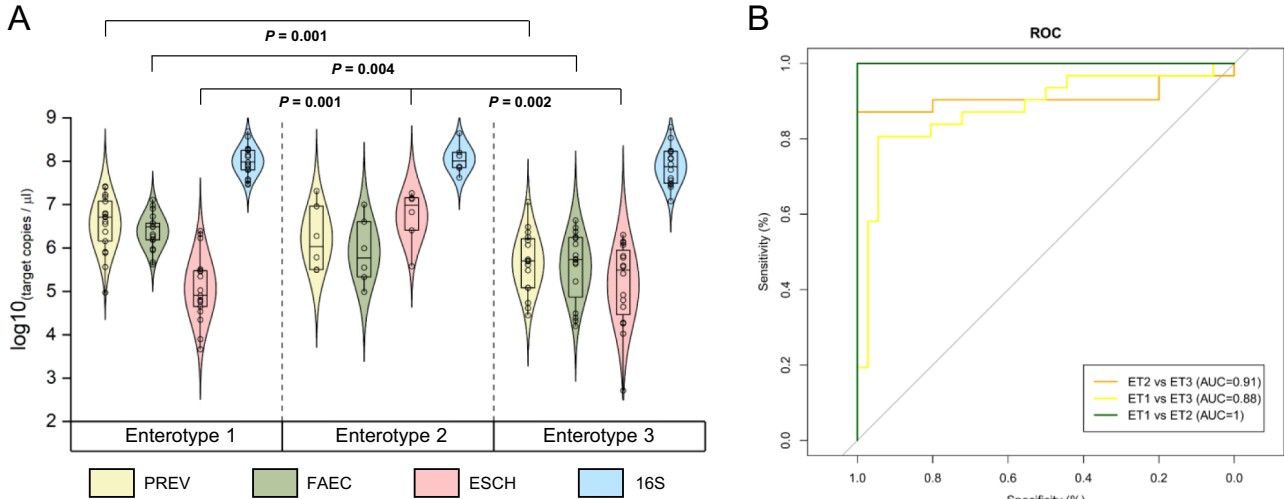

**Fig. 3 Taxon-specific and total bacterial qPCR to classify pre-treatment sample in a treatment-relevant category. A** Total bacteria and taxon-specific density measured by quantitative PCR (qPCR), by enterotype. Comparison of absolute abundances between enterotype 1 ($n = 17$), enterotype 2 ($n = 6$), and enterotype 3 ($n = 16$) was conducted using two-sided Mann-Whitney tests. The lower and upper bound of each box represent the 25th and 75th percentiles, respectively, and the line within indicates the median. The whiskers represent the minimum and maximum values. **B** Classification sensitivity and specificity into the enterotype using qPCR values. PREV *Prevotella* genus qPCR target, FAEC *Faecalibacterium* genus qPCR target, ESCH *Escherichia coli* qPCR target; *16S* total bacteria qPCR target.

the highest CR of 78.6% at 14-days post-treatment whereas patients with ET1 and ET2 reached only 31.3% and 16.6% cure in this time frame, respectively. Daily egg patterns also largely reflect these findings (Fig. 5B). A sharp decrease of EPG counts is observed immediately following treatment, irrespective of treatment arm or baseline enterotype. For *T. trichiura* EPG counts, this decrease was more pronounced in the combination therapy arm (−98.2% average EPG at day 5) than the monotherapy arm (−72.9% average EPG at day 5) which correlates with increased efficacy observed for the combination therapy at the recommended follow-up examination (days 14–28). Notably, EPG counts increase after the initial drop for patients with ET1, and to a lesser extent, patients with ET2, for both types of infections. There was no difference when comparing EPG count decrease for hookworms between treatment arms with an average decrease of 99.1% and 99.3% at day 5 for the combination therapy and monotherapy, respectively.

Functional profiling of the three enterotypes using KEGG orthology terms (KO) showed that a large proportion of KO were shared between the three enterotypes ($n = 3823$ or 80.8%; Fig. 6A). ET3 had the highest proportion of unique KO (4.4%), followed by ET1 (2.6%), and ET2 (1.4%) indicating the highest functional diversity for ET3 and lowest for the *E. coli* dominated ET2. Quantitatively, however, variation in copies per million per KO was higher between ET3 and ET2 (27.2% of KO significantly different) than between ET3 and ET1 (0.96% of KO) indicating functional redundancy between the latter despite pronounced taxonomic differences (Fig. 6B). Significant differences were observed at the community level between the different enterotypes when comparing them using a PERMANOVA (Fig. 6C). However, the difference was much more pronounced when comparing ET2 to ET1 and ET3 with $R^2$ values of 0.338 ($P = 0.001$) and 0.347 ($P = 0.001$), respectively, than between failure-associated ET1 and success-associated ET3 ($R^2 = 0.073$, $P = 0.009$). In Fig. 6D, we show that high-level community metabolic potential is similar between enterotypes and KO involved in genetic information processing, metabolism, as well as signaling and cellular processes account for over 80% of the identified pathways for all enterotypes. We observed significant differences between enterotypes in specific pathways that were

previously linked to positive STH infection status in the literature[22–24]. For instance, three pathways including glycine, serine, and threonine metabolism, prokaryotic carbon fixation, as well arachidonic acid metabolism were significantly enriched in failure-associated ET1 and ET2 when compared to ET3 (Fig. 6E, a complete list of differentially abundant pathways is available in Supplementary Fig. 7).

## Discussion

This study is the first to investigate gut microbial determinants of treatment failure for essential drugs used to treat human *T. trichiura* and hookworm infections. Our results represent a significant improvement in our understanding of gut microbiota-drug interaction in the context of anthelmintic treatment. The key findings are (i) that taxonomically distinct communities were found in this setting in terms of both, relative and absolute abundance of specific taxa, and (ii) that pre-treatment community composition, or enterotype, is associated with treatment outcome of the combination therapy, albendazole, and ivermectin, for *T. trichiura* as well as for hookworm infections. These observations indicate that the pre-treatment microbial composition of stool samples is strongly correlated with the treatment efficacy of both *T. trichiura* and hookworms when using ivermectin-based treatment.

Ivermectin is a functionally and chemically complex drug. For instance, recent literature described the re-positioning of the drug to treat a wide range of off-label non-parasitic diseases[25,26]. As consequence, this could lead to an increase in ivermectin consumption worldwide de facto making understanding of its interactions with the gut microbiota even more relevant for public health. Furthermore, it harbors a 16-membered macrocyclic lactone group, which is structurally similar to that found in important macrolide antibiotics including, erythromycin (14-membered lactone), azithromycin (15-membered lactone), and tyrosin/josamycin (16-membered lactone), among others[27–30]. While this remains to be functionally validated, it raises the question as to whether drug resistance interactions – potentially drug inactivation—could occur between bacteria and ivermectin, which could be driven through a potential bacteriostatic/

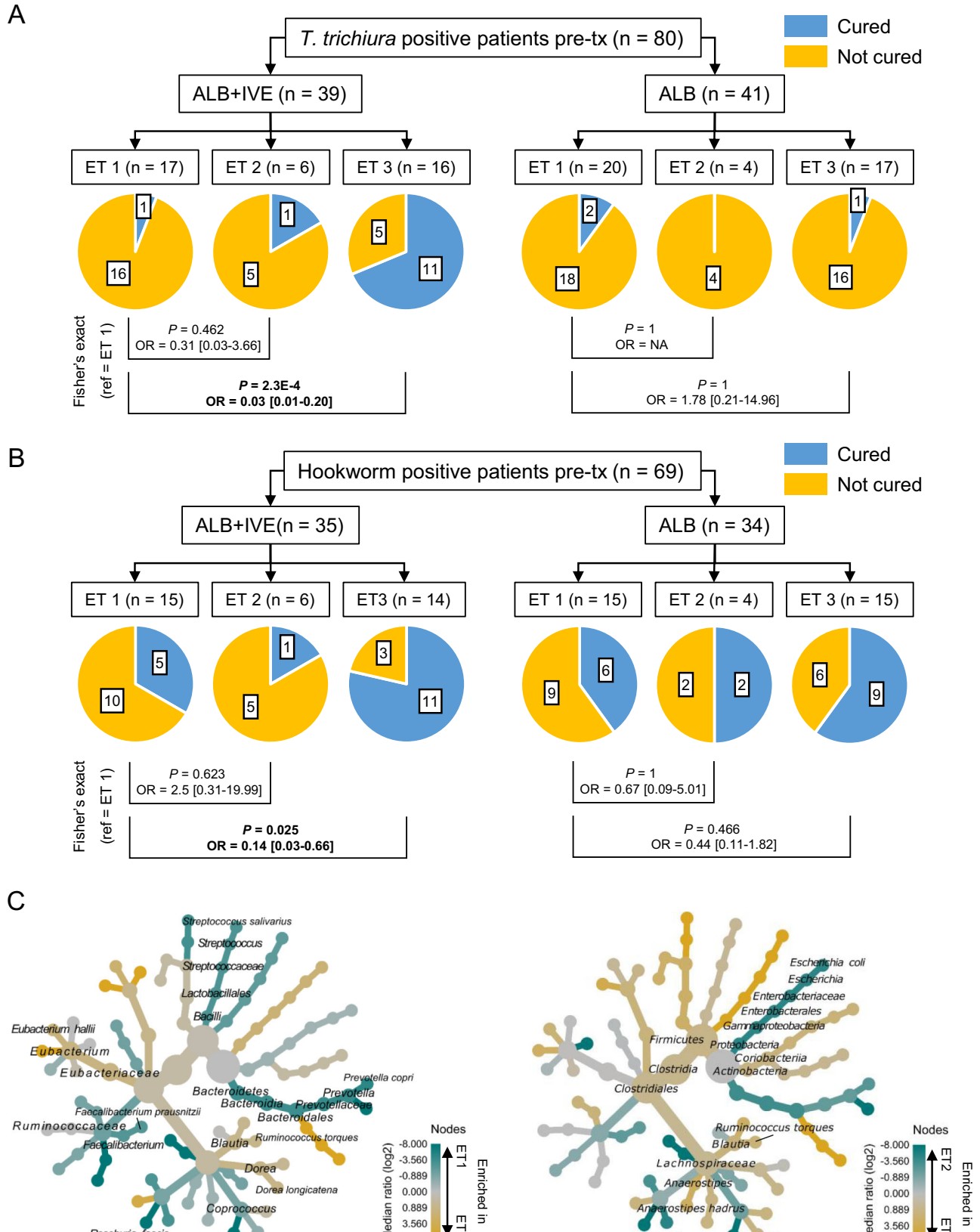

**Fig. 4 Association between soil-transmitted helminths cure and pre-treatment enterotype (ET) by treatment arm and species-level characteristics of each enterotype. A** Association between treatment outcome of *Trichuris trichiura* and ET at baseline. **B** Association measured between treatment outcome of hookworm and ET before treatment. The cure rate is defined as the presence or absence of eggs in stool between day 14 and day 28 after treatment (= average eggs per gram of stool in samples collected between days 14 and 28 after treatment). **C** Species-level differences between compositional clusters (= enterotypes). Fisher's exact tests were two-sided and the 95% confidence interval of the odds ratios is shown in the bracket. Labels on the pie charts represent the number of patients in each group. *n* number of patients, OR odds ratios.

**Table 2 Prediction of time to egg clearance using pre-treatment (pre-T$_x$) enterotypes (ET).**

| Outcome | Predictor | HR (95% CI)—ET pre-T$_x$ | P-value | Predictor | HR (95% CI)—ET pre-T$_x$ | P-value |
|---|---|---|---|---|---|---|
| *Trichuris trichiura*—ALB-IVE | | | | | | |
| Time to egg clearance | ET 2 | 2.94 (0.18–47.06) | 0.445 | ET 3 | 18.12 (2.33–141.04) | 0.005 |
| (ref = ET1) | Adjusted for | | | Adjusted for | | |
| ALB + IVE | Sex | 4.47 (0.27–74.91) | 0.298 | Sex | 29.96 (3.48–258.04) | 0.001 |
| | Age | 3.34 (0.19–56.35 | 0.403 | Age | 20.01 (2.46–162.89) | 0.005 |
| | Infection intensity | 2.39 (0.15–38.43) | 0.536 | Infection intensity | 18.69 (2.39–145.98) | 0.005 |
| | Baseline EPG | 2.49 (0.16–40.24) | 0.518 | Baseline EPG | 17.96 (2.3–139.74) | 0.006 |
| *n* = 39, events = 13 | | | | | | |
| *Trichuris trichiura*—ALB | | | | | | |
| Time to egg clearance | ET 2 | Not applicable (no patient with ET2 was cured) | | ET 3 | 0.61 (0.05–6.69) | 0.683 |
| (ref = ET1) | Adjusted for | | | Adjusted for | | |
| ALB | Sex | | | Sex | 0.6 (0.05–6.64) | 0.678 |
| | Age | | | Age | 0.99 (0.04–22.11) | 0.997 |
| | Infection intensity | | | Infection intensity | 0.61 (0.06–6.75) | 0.688 |
| | Baseline EPG | | | Baseline EPG | 0.65 (0.06–7.21) | 0.723 |
| *n* = 41, events = 3 | | | | | | |
| *Hookworm*—ALB + IVE | | | | | | |
| Time to egg clearance | ET 2 | 0.53 (0.06–4.56) | 0.565 | ET 3 | 3.75 (1.28–10.98) | 0.016 |
| (ref = ET1) | Adjusted for | | | Adjusted for | | |
| | Sex | 0.49 (0.06–4.43) | 0.529 | Sex | 3.45 (1.06–11.28) | 0.04 |
| | Age | 1.05 (0.11–10.2) | 0.969 | Age | 5.67 (1.76–18.28) | 0.003 |
| | Infection intensity | 0.56 (0.07–4.83) | 0.599 | Infection intensity | 3.71 (1.26–10.91) | 0.017 |
| | Baseline EPG | 0.48 (0.06–4.13) | 0.505 | Baseline EPG | 3.1 (1.06–9.12) | 0.039 |
| *n* = 35, events = 17 | | | | | | |
| *Hookworm*—ALB | | | | | | |
| Time to egg clearance | ET 2 | 1.55 (0.31–7.71) | 0.592 | ET 3 | 1.69 (0.59–4.75) | 0.322 |
| (ref = ET1) | Adjusted for | | | Adjusted for | | |
| | Sex | 1.57 (0.31–7.81) | 0.584 | Sex | 1.88 (0.63–5.59) | 0.257 |
| | Age | 1.62 (0.29–8.7) | 0.577 | Age | 1.78 (0.52–6.05) | 0.357 |
| | Infection intensity | 1.55 (0.31–7.75) | 0.591 | Infection intensity | 1.67 (0.59–4.7) | 0.332 |
| | Baseline EPG | 1.72 (0.34–8.72) | 0.514 | Baseline EPG | 1.64 (0.58–4.61) | 0.350 |
| *n* = 34, events = 23 | | | | | | |

Clearance is defined as two consecutive samples with no eggs (*Trichuris trichiura* or hookworm) detected and an average egg count between days 14 and 28 equal to zero. A significant hazard ratio above one indicates that the patient is reaching the event (clearance of eggs) at a faster rate. *ALB-IVE* albendazole and Ivermectin, *ALB* albendazole, *ref.* reference category, *HR (95% CI)* hazard ratio (95% confidence interval), *T$_x$* treatment.

bactericidal activity of the drug on failure-associated taxa. Further studies should investigate this direct mechanism since several of the failure-associated taxa are gram-positive bacteria, which are known targets of macrolide antibiotics and could hence be sensitive to this drug[29]. Drug resistance isn't the only possibility, and a mechanism recently described in *S. salivarius*[31]—the biosequestration of drugs in bacterial cells, could also contribute to decreased treatment efficacy. Finally, another potential mechanism of interaction could be driven by the metabolic potential at the community level. Currently, data about interactions between the gut microbiota and helminth colonization is rather scarce—the main source being observational cohorts comparing the metabolic make-up of infected and non-infected patients using high-throughput sequencing approaches. In this study, we highlighted community-specific differences in the abundance of metabolic pathways which—in turn—could potentially be linked to treatment outcomes. While assessing the causal link will require subsequent studies, we show that the metabolic potentials of the three treatment-relevant enterotypes identified in this study present significant differences. This is especially true for failure-associated ET2 which presented profound differences with the two other enterotypes, ET1 and ET3. In addition, we found several specific pathways to be enriched in both failure-associated

phenotypes—when compared to ET3—that were previously associated with positive STH infection status. While this remains to be validated, we could hypothesize that certain microbial compositions—and the associated metabolic traits—could provide a more or less permissive environment for the colonizing parasites and result in modulated efficacy of anthelminthic treatments. It is worth noting that this microbiome-drug interaction is present for hookworms, which colonize the small intestine, as well as for *T. trichiura*, which resides in the cecum, at the beginning of the large intestine. This speaks in favor of a more widespread interaction, rather than a locally restricted event. Another key focus point of future research would also include understanding how many, among the 34–86% of non-responders to albendazole/ivermectin-based treatment, can be attributed to potential bacteria-driven interactions. This information would help define how treatment response could be improved through microbiome-based therapy.

The other important implication of this study is that ivermectin, which is marketed as an antiparasitic drug, might also contribute to selecting cross-resistant bacterial isolates, which could hinder efforts involved in antibiotic stewardship programs. This is particularly important since it is widely used in MDA campaigns. Provided this resistance interaction is functionally

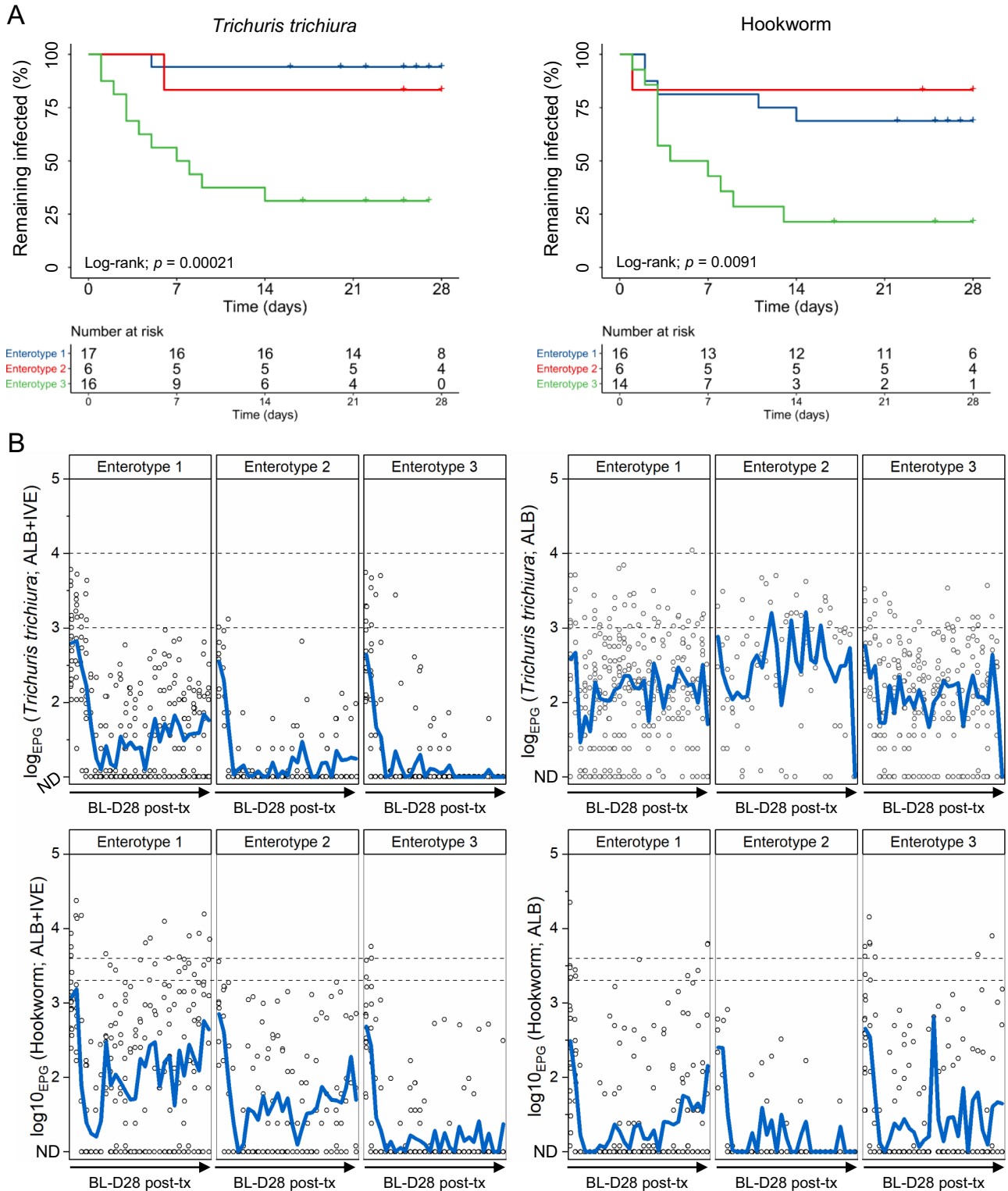

**Fig. 5 Relationship between pre-treatment enterotype and daily post-treatment egg counts of *Trichuris trichiura* and hookworm. A** Kaplan–Meier curve showing the remaining proportion of infected patients over time, stratified by baseline enterotype. *T. trichiura* cure is shown on the left, while hookworm cure is shown on the right. **B** Averaged daily egg counts of *T. trichiura* (upper two panels) and hookworm (lower two panels) stratified by pre-treatment enterotype. Daily egg counts are shown for the combination therapy (albendazole and ivermectin; left half panels) and monotherapy (albendazole; right half panels). The black dashed lines indicate the lower and upper thresholds of moderate infections for each parasite. ND not detected, post-tx post-treatment, ALB albendazole, IVE ivermectin.

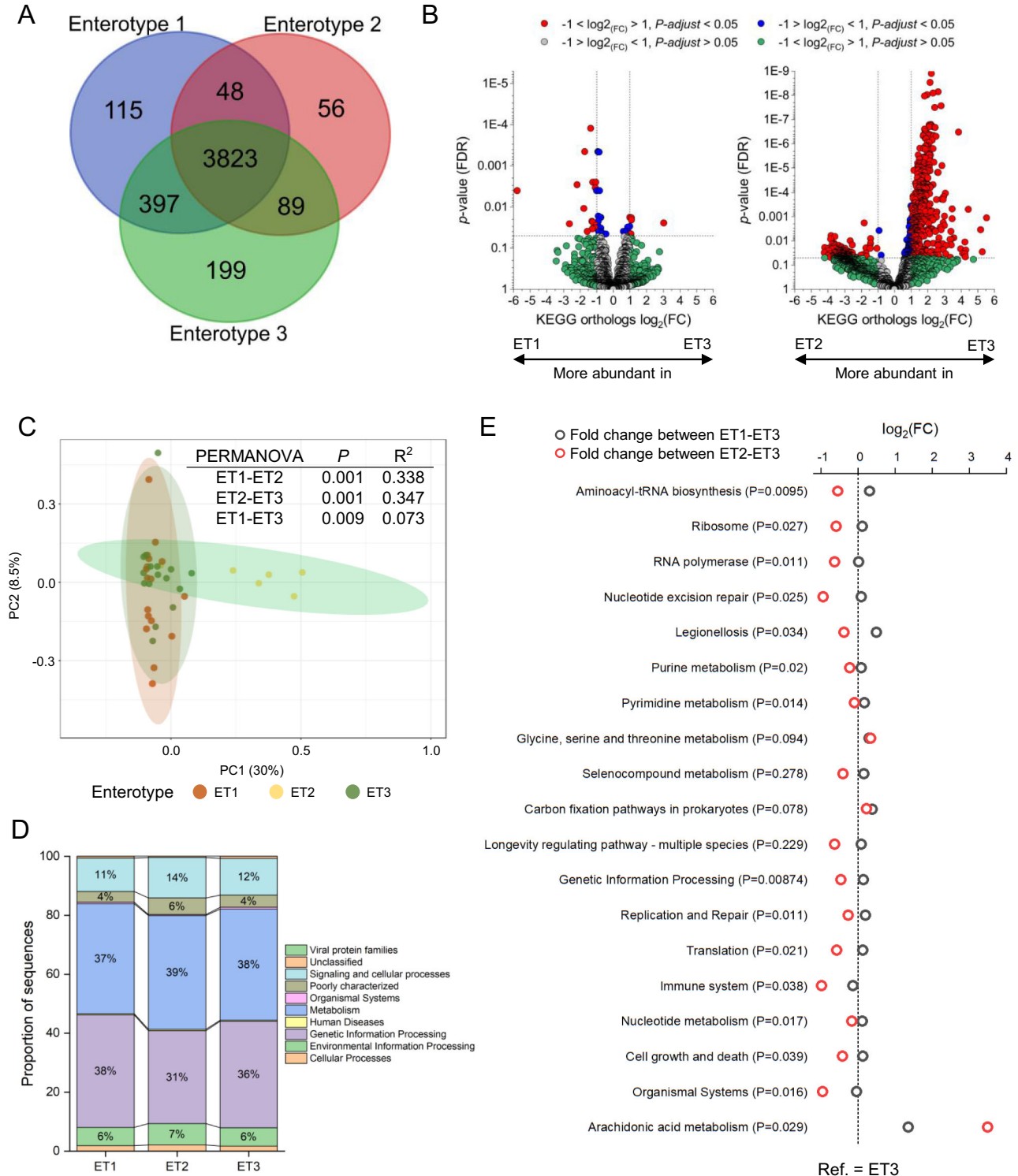

**Fig. 6 Comparison of metabolic potential between enterotypes. A** Venn diagram showing qualitative overlap of KEGG orthology terms (KO) between treatment-relevant enterotypes. **B** Volcano plots highlight quantitative differences (differences in normalized counts) at the KO level between enterotypes. The comparison was performed using Deseq2 and $P$-values were adjusted for multiple testing bias using the Benjamini–Hochberg procedure. **C** Principal component analysis to display community-level differences of KO. A PERMANOVA analysis was conducted to quantify these differences. **D** Bar chart showing the average relative abundances of high-level KEGG pathways, by enterotype. For clarity, only proportions of sequences above 2% are shown. **E** Comparison between enterotypes of relative abundances of metabolic pathways associated with positive soil-transmitted helminth infection status in previous studies[22–24]. A Kruskal–Wallis test was used in combination with the Benjamini–Hochberg multiple testing bias corrections to identify differences and fold changes were calculated to show the corresponding effect sizes. ET1–3 enterotypes 1–3, FC fold change, Ref. Reference.

validated, ivermectin and a related derivative moxidectin should also be considered in antimicrobial stewardship efforts. This could be achieved by using pre-treatment screening of genetic/ phenotypic determinants of resistance.

However, our study has several important limitations. First, our observations are based on a cohort with a relatively low sample size ($n = 80$) and on the disputed concept of enterotypes. In practice, this means that the enterotype classification based on beta-diversity that we propose in this study might change in future studies. In addition, the taxonomic determinants of failure identified in this setting might be specific to the village of PakKhan in Laos and might not be generalizable. While we confirmed the robustness of our taxonomic-related observations with different techniques, further studies in different settings are needed to pinpoint common microbial/genetic denominators of low albendazole plus ivermectin-based treatment efficacy. Second, we did not investigate intrinsic worm resistance. While there is currently no known target for screening of *T. trichiura* and hookworm resistance to ivermectin, we cannot exclude that lower sensitivity of the parasites might explain, in part, decreased treatment efficacy. While this could be the case, our efficacy-related observations (66.7% failure) are in line with previous observations of failure associated with this drug combination. Since there is no evidence of parasitic resistance to ivermectin in human populations to date, we believe that the study setting is representative of an environment with normal parasitic sensitivity.

In conclusion, we showed in this study that the gut microbiota composition is an important driver of response to anthelminthic treatment based on albendazole and ivermectin. These findings will enable understanding of the cause of albendazole and ivermectin-based treatment failure in a large proportion of patients, while still being one of the most efficacious options in terms of success to treat STH infections. This will hopefully lead to novel therapeutic opportunities based on the modulation of failure-associated features, and—perhaps more importantly—to optimized, evidence-based use of these powerful drugs.

## Methods

**Sample collection, data collection, and microscopy.** Stool samples were collected within the framework of a multi-country randomized controlled trial assessing the efficacy and safety of the drug combination albendazole-ivermectin against *T. trichiura* and concomitant helminth infections[12]. *T. trichiura* infection status assessed by Kato-Katz 14–21 days after treatment was the primary endpoint of the trial and the main outcome for efficacy was expressed as CR (i.e., conversion from being egg positive pre-treatment to egg negative post-treatment) and ERR (secondary endpoint). Secondary endpoints included further infection status with *A. lumbricoides*, hookworm, and *S. stercoralis* and related efficacy measures and adverse events. In the Nambak District where the microbiome sub-study was conducted, the village of Pak-Khan was selected for the daily sample collection for 28 days post-treatment based on previous good compliance and an adequate number of participants living there. In detail, individuals (including parents/ caregivers of children) interested in participating in the trial were invited to complete the process of informed consent; thereafter, individuals were assessed for study eligibility during screening procedures. Prior to the start of collection, participants were informed of the aim of the daily sample collection, in addition to the consenting and information sessions conducted for the trial[32]. For screening, two stool samples from different days (within a 5-day interval) were collected and transported to the Nambak hospital. For each stool specimen, duplicate Kato-Katz thick smears (41.7 mg each) were prepared and read under a microscope for eggs of *T. trichiura*, *A. lumbricoides*, hookworm, and *O. viverrini* by experienced technicians. Participants underwent a pre-treatment physical examination to collect clinical data and treatment took place approximately 1 week after baseline screening[32]. A small aliquot of <1 g stool was transferred to a 2 ml screwcap cryotube using a UV-sterilized plastic spatula and immediately frozen at −20 °C. After the conclusion of the respective trial stage, the frozen samples were shipped directly to Swiss TPH, Basel, Switzerland on dry ice and kept at −20 °C until analysis.

**DNA isolation.** Samples were extracted using QIAamp DNA Mini kit (Qiagen, Hilden, Germany). The protocol used a garnet bead-beating approach in reference

to a standard protocol developed by Kaisar et al.[33] with minor modifications. It has been described in more detail elsewhere[34].

**16S rRNA gene and shotgun sequencing.** The V3–V4 hypervariable region of the 16S rRNA gene was sequenced as described previously[35]. Briefly, amplification reactions were performed using 12.5 μl of Kapa HiFi HotStart ReadyMix (Roche, Basel, Switzerland) mix, 1.5 μl of 10 μM forward and reverse primers, 7.5 μl of sterile water, and 2 μl of template DNA. The V3–V4 region was amplified by cycling the reaction at 95 °C for 3 min, 30× cycles of 95 °C for 15 s, 50 °C for 15 s and 72 °C for 15 s, followed by a 5 min 72 °C extension. To avoid bias associated with PCR amplification, all reactions were done in triplicate, visually controlled on a 1.2% agarose and subsequently pooled together. Each pool was quantified using a Qubit hsDNA assay (Thermo Fisher Scientific, Waltham, MA, USA). The purified library was loaded onto an Illumina Miseq sequencer (Illumina, San Diego, CA, USA) as recommended by the manufacturer. Sequencing was performed using V3 chemistry (600 cycles) in 2× 300 bp mode. Shallow shotgun sequencing was conducted as described previously[36]. Briefly, we generated 4.4–13.3 M reads per sample to obtain representative taxonomic profiling with a better taxonomic resolution than what is allowed with 16S rRNA gene sequencing. DNA concentration was measured from 2 μl isolated DNA using a Qubit 4.0 in combination with hsDNA quantification kits (Thermo Fisher Scientific, Waltham, MA, USA). Libraries were prepared using the NEBNext Ultra II FS DNA kits (New England Biolabs, Ipswich, MA, USA) according to the manufacturer's protocol. Final libraries were quantified and pooled together in an equimolar mix. The pool was subsequently loaded onto an Illumina NextSeq sequencer and sequenced using a Mid-output kit (300 cycles) in paired-end mode (2× 150 bp). Sample sequencing depth is available in Supplementary Table 1.

**Taxonomic and functional profiling.** The QIIME 2 pipeline v. 2020.8[37] was used to analyze data generated by 16S rRNA gene sequencing. Briefly, after data import, Deblur was used to correct reads and cluster them into amplicon sequence variants with the option "--p-trim-length 140". Taxonomic classification was done using the 16S rRNA gene database SILVA v.138[38]. Mitochondria and chloroplast-related ASVs were removed using the "taxa filter-table" command. Alpha rarefaction curves were generated using the "diversity alpha-rarefaction" command with "–p-max-depth 25000" option. Phylogenetic distances were calculated using the core-metrics-phylogenetic script and samples below 3000 classified reads were excluded using the option "–p-sampling-depth 3000". Cleaned tables were exported and converted using the BIOM tool suite v. 2.1.9[39]. Metaphlan v. 3.0.7[40] was used to perform taxonomic profiling on sequence data generated using the shallow shotgun sequencing technique. The species-markers database was CHOCOPhlAn v30 dating from January 2019 and reference mapping was done using Bowtie2 v. 2.4.5[41]. The option "–add_viruses" was added to the Metaphlan command to allow taxonomic profiling of viruses. Taxonomic profiles generated with Metaphlan3 were normalized by 16S qPCR values. A low count filter (minimum count = 100; prevalence = 20% of samples) and a low variance filter based on the interquartile range (50% features removed) were applied to the count table. Finally, data were transformed using the centered log-ratio method[42] before statistical analysis.

Functional profiles were generated with Humann3 v. 3.0.0.alpha.3[40] using default options in combination with the included UniProt/Uniref 2019_01 database. The resulting gene abundance tables were converted to a KO table using the humann_rename_table.py script. Upon renaming, tables were re-normalized using the humann_renorm_table.py script. Subsequently, KO terms were regrouped into KEGG pathways using a custom KEGG pathway definition file. Finally, differential analysis of KO was conducted using Deseq2 v. 1.34.0[43], and group comparison was estimated using a custom fold-change script in combination with Kruskal–Wallis tests adjusted for multiple testing bias using the Benjamini-Hochberg procedure[44].

Effect of sequencing depth on taxonomy-related metrics (e.g., alpha diversity indices) and functional metrics (e.g., number of identified KO or number of reconstructed pathways) was assessed using Spearman correlation as shown in Supplementary Table 2. We did not observe bias due to sequencing depth for any metric.

**qPCR analyses.** qPCR assays were conducted on a CFX Opus real-time PCR system (Bio-Rad, Cressier, Switzerland) using Taqman Gene Expression analyses kits (Thermo Fisher Scientific, Waltham, MA, USA). qPCR targets included total bacteria (targeting the 16S gene)[45], *Prevotella* genus[46], *Faecalibacterium prausnitzii*[47], and *Escherichia coli*[48]. Briefly, DNA was first diluted 10× in ultrapure water and each 10 μL reaction contained 0.3 μM of forwarding primer, 0.3 μM of reverse primer, and 0.2 μM (0.1 μM for *E. coli*) of Taqman probe, 5 μL of master mix, 2.2 μl of ultrapure water, and 2 μL of DNA template. Cycling conditions were as follows: 10 min at 95 °C followed by 40 cycles at 95 °C for 15 s and 60 °C for 1 min. The detection threshold was set at 1500 RFU for targets detected with FAM, 700 RFU for HEX, and 200 RFU for CY5, and each reaction was conducted in duplicate. Primers and probes are summarised in Supplementary Table 3. Data analysis of the raw qPCR signal was performed using the CFX Maestro Software 2.0 v. 5.0.21.0616 (Bio-Rad, Cressier, Switzerland).

**Statistical analysis**. Group comparison using Mann–Whitney's test, Fisher's exact test of independence, and Spearman correlation tests were performed using XLSTAT v2020.2.3 (Addinsoft, New York, USA). Enterotype classification is derived from the taxonomic profiles obtained by 16S rRNA gene sequencing. Classification is based on a Bray–Curtis dissimilarity matrix and the dendrogram was obtained using Ward's algorithm. A $k$-means clustering approach was used to define the optimal number of clusters which was set to 3 (Supplementary Fig. 8). The cladogram was then divided using the "cutree" function from the hclust package with the option "$k = 3$" which subsequently resulted in the annotation of each sample with the enterotype they belong to. Beta diversity measures and NMDS ordination plot derived from 16S rRNA gene sequencing were calculated in R v3.6.3[49] with the following packages: vegan v2.5–6[50] (with the vegdist and metaMDS functions) and hclust[51]. The package randomForest v4.6–14[52] was used to run random forest models on 16S sequencing and qPCR data. The receiver operating characteristic (ROC) curves and AUC calculations were done using the pROC package v.1.16.2[53]. To identify enriched features in each enterotype, we used the LefSe package v.1.0[54] (available on https://huttenhower.sph.harvard.edu/galaxy/). The survival analysis and Cox proportional hazard were generated and measured using the "survfit" and "coxph"[55] functions from the survival packages, respectively. The heat tree analysis was performed on the MicrobiomeAnalyst platform (06/01/2021)[56] using Metacoder R package v. 0.3.5.001[57]. Relative abundances of enterotype-relevant taxa at the genus level were correlated using Spearman correlation to assess agreement between the different detection techniques (16S rRNA gene sequencing, shallow shotgun sequencing, and the qPCR-derived ratio of target density over total bacteria density (Supplementary tables 4 and 5). All graphs, besides the heat tree, were generated using the OriginPro 2021 graphing software v9.8.0.200 (OriginLab Corporation, Northampton, MA, USA).

**Ethical approval and consent to participate**. The trial was approved by the Ethics Committee of Northwestern and Central Switzerland (EKNZ; BASEC Nr Req-2018-00494) and by the National Ethics Committee for Health Research in Lao PDR (No 093/NECHR). This trial was registered under the number NCT03527732 on ClinicalTrials.gov. Informed consent was obtained for all participants. No compensation was offered to participate.

**Reporting summary**. Further information on research design is available in the Nature Research Reporting Summary linked to this article.

## Data availability

The sequencing data (16S rRNA gene sequencing and shotgun sequencing) generated in this study have been deposited in the NCBI Short Read Archive under accession code PRJNA767599.

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

## Acknowledgements
We are indebted to all local helpers including the village and medical authorities of the Luang Prabang province in Laos. We thank Christian Beisel and Ina Nissen from the Genomics Facility Basel for their support during the shotgun sequencing experiments. Calculations were performed at sciCORE (https://scicore.unibas.ch) scientific computing center at the University of Basel on 40 cores and 480GB RAM. We thank Kateryna Skrypko for proofreading the paper. We are grateful to the European Research Council (No. 101019223) for financial support.

## Author contributions
P.H.H.S.: study design, research design, project supervision, experimental work, statistical analyses, figure generation, writing of the initial paper, and paper editing; M.G.: experimental work (16S rRNA gene sequencing, manuscript editing; S.W., E.H., and S.S.: study design, conducted fieldwork (sample collection, handling, coordination of treatment, parasitological work, and data curation), experimental work (S.W.; DNA isolation), paper editing; J.D. and C.H.: experimental work (sample handling, DNA isolation, qPCR), paper editing; J.E.F.: experimental work (supervision of sequencing), paper editing; J.K.: study design, research design, project supervision, writing of the initial paper, paper editing.

## Competing interests
The authors declare no competing interests.
