## [Peer Review File · Nature Communications]

Different gut microbial communities correlate with efficacy of albendazole-ivermectin against soil-transmitted helminthiasesReviewers' Comments:

Reviewer #1:

Remarks to the Author:

The current study explores the potential that different gut bacterial enterotypes may impact on the efficacy of anthelmintic treatment, specifically looking at the association between gut enterotype and success in treating hookworm or whipworm infections. Broadly speaking the work is sound and the methods applied appear appropriate. I think the results of the study justify the primary conclusions made by the authors and I do not think they have over-interpreted their study.

However, although the work is interesting, unfortunately, I do not feel it is of sufficiently high impact for Nature Communications and belongs in a parasitology or tropical medicine specific journal.

Firstly, as the authors note the study is small and has a number of limitations. This could be overlooked if it was describing a finding that had the potential to drastically improve treatment in the near future. However, I just don't see a link between these findings and any clear change to deworming practices in areas that are significantly impacted by STH infections. STH infections are a significant issue in low-income countries and highly impoverished populations. Treatment is largely based on mass deworming programs that deliver, primarily, oral benzimidazoles, to tens of millions of school-aged children once per year at a cost of about \$0.03 USD per dose. I can't imagine any scenario in which large-scale microbiomic screening programs to identify enterotypes likely to be amenable to treatment would be implemented. At present, in most countries, mass deworming programs rarely test for infection prior to treatment, as this is simply too costly and impractical at the scale needed to impact on STH burden.

That doesn't mean that I think the work is not interesting. It is, but it is interesting within the context of informing pharmacology and pointing toward a potentially important observation that, as yet, does not have a known mechanism to explain it. This is perfectly publishable as a preliminary study in a discipline-specific journal and would likely find plenty of attention there.

Reviewer #2:

Remarks to the Author:

In this manuscript by Pierre et.al., the investigators investigated the relationship between the gut microbiota and anthelmintic treatment failure for human *T. trichiura* and hookworm infections. Whereas anthelmintics have always been very effective against hookworm and *Ascaris*, there is considerable heterogeneity in *Trichuris* responses to treatment, which is poorly understood. This manuscript provides the first insight into gut microbiota-drug interactions in the context of anthelmintic treatment. Stool samples were collected pre-post treatment in Laos for 80 patients for sequencing and determination of treatment efficacy. By sequencing, they found that the samples fell into 3 enterotypes. What was most interesting is that cure rate for enterotype 3 was higher than ET1 and ET2. Although the concept of enterotypes can be controversial, the data presented here does look quite convincing, although it is not clear to me if the definition is by 16S or metagenomics (see below). To my knowledge, this is the first time deworming efficacy has been clearly linked to the microbiota composition pre-treatment, indicating a possibility of affecting drug metabolism. Hence, this is a substantial contribution to the field. One of the strengths of this manuscript is that the authors have taken very complex data (which could be analyzed in a large number of different ways) and distilled an interesting and important message from their analysis that is clear and well presented. However, the manuscript could potentially be improved by a few additional considerations described below.

1. Its not entirely clear to me how the species composition is determined? By 16S or shotgun or both, if both, how is the data combined? Do the shotgun metagenomic data also group the samples into

enterotypes? Is what is shown for 16S? it's possible that the large number of unmapped reads that can arise from shotgun sequencing may alter these classifications. There needs to be greater clarification of how microbial community composition is defined, by 16S or shotgun, etc.. as this could affect the enterotype classification?

2. One advantage of having metagenomic sequencing data is that there is the possibility of assembling metabolic pathways and microbial functions within the bacterial communities that may provide some mechanistic insights into why ET3 can achieve higher cure rates. This does not appear to have been attempted by the authors. It could be that the sequencing depth is too shallow etc... but the authors should nonetheless comment on this.

3. The authors have broken down the data into purely categorical variables (cure vs non-cure). However, egg burden is a continuous variable and there is variation in responses that are more complex than cure vs non-cure. For example, "change in egg burden" i.e. there will be individuals where egg burden is reduced but not cure and others that are not reduced. Why would enterotypes not be associated with change in egg burden, but with cure vs non-cure?. Some discussion of this could be important.

4. It is a bit of a puzzle to me why the Hookworm cure rate is so low in this community. Most studies, as well as our own experience, typically describe much higher cure rates closer to 90-100%. The authors may want to provide some insight and discuss this in their discussion.

5. It should be noted that ET3 is the most diverse of the enterotypes, the authors may want to comment on why the more diverse the bacterial community, the better they seem to achieve higher cure rates.

MINOR COMMENTS

Results:

1. Line 120: "Among these, 86.2% (n=69) were also co-infected with hookworms, 40% 120 (n=32) with *A. lumbricoides*, and 8.7% (N=7) with *Opisthorchis viverrini*." Are there samples with only one infection?

Methods:

1. Line 342-343: Author mentioned "The V3-V4 hypervariable region of the 342 16S rRNA gene was sequenced as described previously", but in line 346 "The V4 region was amplified by cycling the reaction..." . Please edit.

Reviewer #3:

Remarks to the Author:

The research paper by Scheenberger et al approaches a relevant question in antihelmintic therapy and contributes with meaningful methodologic tools for the understanding of the variable and overall low cure rates of infections due to some species of STH. The establishment of links between treatment response and gut microbium triggers very important hypothesis, and this report does it.

In order to make adequate interpretation of these results and identify follow-up questions, some clarifications are needed to fully support its conclusions.

Major comments:

The main analysis is based on 3 enterotypes; however, there is no clarity whether these is a pre-defined analysis or rather a post-hoc analysis based on the selected enterotypes when the results of cure rates were already available. The rules and criteria used for defining the 3 groups instead of other groups and/or a different number of enterotype groups needs to be clearly defined; also, whether the definition of these groups was done blinded of the treatment response results; along the same lines, the accuracy that is referred in line 134 should be further clarified.

Although in line 204 it is stated that the findings remain after adjusting for infection intensity, a more

robust adjustment on the same variable could be based on baseline egg count and its correlation to enterotype; perhaps a graph and analysis similar to what it is applied in Figure 3 of the supplement could be useful.

The definition of "clearance" (and cure?) is stated as "two consecutive samples with no eggs (*Trichuris trichiura* or hookworm) detected and an average egg count between day 14 and 28 equal to zero" is confusing and hard to understand, since the finding of any amount of eggs equal or higher than 1 in any post treatment sample from day-14 on, should be considered as a treatment failure.

Minor comments

The abstract should include the sample size of each experimental group.

Ln75. The word "advertise" is probably inadequate for the intended meaning of that sentence.

Ln94-96. The sentence seems incorrect in its construction.

The long discussion centered on the potential role of ivermectin as a substrate of macrolide-resistance as well as a promoter of drug resistance appears as too much speculative and although a thought-provoking idea, might be soften and supported by empirical data.

Point-by-point response:

We thank the editor and the reviewers for providing the opportunity to revise and resubmit our manuscript. We are confident that the overall quality of our manuscript has further improved by addressing the suggestions put forth by all referees. Hence, we hope that it will now be suitable to be considered for publication in *Nature Communications*.

Responses are highlighted in blue.

Reviewer #1 (Remarks to the Author):

The current study explores the potential that different gut bacterial enterotypes may impact on the efficacy of anthelmintic treatment, specifically looking at the association between gut enterotype and success in treating hookworm or whipworm infections. Broadly speaking the work is sound and the methods applied appear appropriate. I think the results of the study justify the primary conclusions made by the authors and I do not think they have over-interpreted their study.

However, although the work is interesting, unfortunately, I do not feel it is of sufficiently high impact for Nature Communications and belongs in a parasitology or tropical medicine specific journal.

Firstly, as the authors note the study is small and has a number of limitations. This could be overlooked if it was describing a finding that had the potential to drastically improve treatment in the near future. However, I just don't see a link between these findings and any clear change to deworming practices in areas that are significantly impacted by STH infections. STH infections are a significant issue in low-income countries and highly impoverished populations. Treatment is largely based on mass deworming programs that deliver, primarily, oral benzimidazoles, to tens of millions of school-aged children once per year at a cost of about \$0.03 USD per dose. I can't imagine any scenario in which large-scale microbiomic screening programs to identify enterotypes likely to be amenable to treatment would be implemented. At present, in most countries, mass deworming programs rarely test for infection prior to treatment, as this is simply too costly and impractical at the scale needed to impact on STH burden.

That doesn't mean that I think the work is not interesting. It is, but it is interesting within the context of informing pharmacology and pointing toward a potentially important observation that, as yet, does not have a known mechanism to explain it. This is perfectly publishable as a preliminary study in a discipline-specific journal and would likely find plenty of attention there.

We thank the reviewer for the positive overall appraisal of our study. We agree with the fact that bringing microbiome profiling in the field in LMICs presents several challenges – the cost being one of them. We are aware that sequencing and subsequent bioinformatics analyses will be difficult to implement in a resource-constrained setting, which is why we explore alternatives (e.g. qPCR) to classify patients into treatment-relevant categories in this study. Further, we feel that understanding the exact mechanisms of interaction will allow translating this research into the field, by pinpointing a single or a restricted set of molecular targets that could – ultimately - be queried on-site using cheaper portable detection methods (e.g. LAMP assays). Moreover, considering the potential impact of this research on the treatment efficacy of these widespread diseases in LMICs, it could be argued that improving the efficacy of treatment has a cost-saving potential in addition to making sure that these essential drugs are used in a more evidence-based rather than prophylactic way. Lastly, as ivermectin is a widely used drug, the understanding of its interactions with the gut microbiota is relevant for many disease areas.

Reviewer #2 (Remarks to the Author):

In this manuscript by Pierre et.al., the investigators investigated the relationship between the gut microbiota and anthelmintic treatment failure for human *T. trichiura* and hookworm infections. Whereas anthelmintics have always been very effective against hookworm and *Ascaris*, there is considerable heterogeneity in *Trichuris* responses to treatment, which is poorly understood. This manuscript provides the first insight into gut microbiota-drug interactions in the context of anthelmintic treatment. Stool samples were collected pre-post treatment in Laos for 80 patients for sequencing and determination of treatment efficacy. By sequencing, they found that the samples fell into 3 enterotypes. What was most interesting is that cure rate for enterotype 3 was higher than ET1 and ET2. Although the concept of enterotypes

can be controversial, the data presented here does look quite convincing, although it is not clear to me if the definition is by 16S or metagenomics (see below). To my knowledge, this is the first time deworming efficacy has been clearly linked to the microbiota composition pre-treatment, indicating a possibility of affecting drug metabolism. Hence, this is a substantial contribution to the field. One of the strengths of this manuscript is that the authors have taken very complex data (which could be analyzed in a large number of different ways) and distilled an interesting and important message from their analysis that is clear and well presented. However, the manuscript could potentially be improved by a few additional considerations described below.

1. Its not entirely clear to me how the species composition is determined? By 16S or shotgun or both, if both, how is the data combined? Do the shotgun metagenomic data also group the samples into enterotypes? Is what is shown for 16S? it's possible that the large number of unmapped reads that can arise from shotgun sequencing may alter these classifications. There needs to be greater clarification of how microbial community composition is defined, by 16S or shotgun, etc.. as this could affect the enterotype classification?

Enterotype classification as described throughout the manuscript was established based on the data generated from 16S rRNA gene sequencing, since all samples – including those from the monotherapy arm – were sequenced using this approach. Shotgun sequencing was conducted solely on samples from the combination therapy arm to identify species-level differences between enterotypes (and potential metabolic pathways associated with treatment outcome). We haven't assessed correlation between taxonomic profiles obtained from the two sequencing technologies – mostly because of taxonomy-related inconsistencies between databases (e.g. ChocoPhlAn versus SILVA) used during profiling. To clarify/remedy these issues in our study, we have modified the manuscript as follows:

a) We have added the following text between lines 449 to 456 to clarify how enterotype classification was obtained:

“Enterotype classification is derived from the taxonomic profiles obtained by 16S rRNA gene sequencing. Classification is based on a Bray-Curtis dissimilarity matrix and the dendrogram was obtained using Ward's algorithm. A k-means

clustering approach was used to define the optimal number of clusters which was set to 3 (**Supplementary figure 8**). The cladogram was then divided using the “cutree” function from the hclust package with the option “k = 3” which subsequently resulted in the annotation of each sample with the enterotype they belong to.”

b) We have added a table (**Supplementary table 4**) to compare relative abundances of the eight genera found to be enriched in either enterotype between sequencing techniques. The correlation was measured using Spearman correlation on pairs of samples sequenced with both, 16S rRNA gene sequencing as well as shotgun sequencing. We have found a strong agreement between both sequencing techniques for 7 out of 8 genera. In addition, we correlated relative abundances from the two sequencing techniques to the relative abundances as measured by qPCR (Target abundance / total bacteria abundance) for the three qPCR targets, *Prevotella*, *Faecalibacterium*, and *Escherichia* as shown in **Supplementary table 5**.

c) We have added the following text at lines 466-470 to describe how we compared the different profiling techniques (16S rRNA gene sequencing, shallow shotgun sequencing, and qPCR analyses):

“Relative abundances of enterotype-relevant taxa at the genus level were correlated using Spearman correlation to assess agreement between the different detection techniques (16S rRNA gene sequencing, shallow shotgun sequencing, and the qPCR-derived ratio of target density over total bacteria density (**Supplementary tables 4 and 5**).”

Finally, we agree that the overall concept of “enterotype” is disputed in the field since community-composition is rather a continuum (as shown in the ordination plot 2D) than a discrete variable. This is the reason why we also added the qPCR-based classification - to provide an initial alternative approach to categorize patients in a sequencing-independent manner. However, we have added the following text (lines 331-333) to the limitation section to make it clear that the enterotypes defined in this study are study-specific and that future research should focus on more specific targets - if possible – rather than overall community composition.

“and on the disputed concept of enterotypes. In practice, this means that the enterotype classification based on beta-diversity that we propose in this study might change in future studies.”

2. One advantage of having metagenomic sequencing data is that there is the possibility of assembling metabolic pathways and microbial functions within the bacterial communities that may provide some mechanistic insights into why ET3 can achieve higher cure rates. This does not appear to have been attempted by the authors. It could be that the sequencing depth is too shallow etc... but the authors should nonetheless comment on this.

We have added supplementary data to assess the effect of low sequencing coverage depth with the shallow shotgun sequencing approach on community metrics. This includes taxonomy-related measures (alpha diversity indices) as well as the number of identified KEGG Orthology (KOs) terms and reconstructed metabolic pathways as a representation of the functional content (**Supplementary table 2**). We did not observe bias for taxonomic features nor for functional features in our dataset. We added the following text at lines 429-434:

“Effect of sequencing depth on taxonomy-related metrics (e.g. alpha diversity indices) and functional metrics (e.g. number of identified KOs or number of reconstructed pathways) was assessed using Spearman correlation as shown in **Supplementary table 2**. We did not assess human DNA content as it usually represents low proportions of the overall content in stool samples⁴⁵. We did not observe bias due to sequencing depth for any metric.”

Hence, we have also added a functional comparison between enterotypes at the community level. We identified differences between enterotypes in terms of KOs presence as well as in terms of KOs abundance (**Figure 6A-B**). We also put these results in the context of which metabolic pathways they are involved in. These results are now summarized from lines 246 to 268:

“Functional profiling of the three enterotypes using KEGG orthology terms (KOs) showed that a large proportion of KOs were shared between the three enterotypes (n = 3823 or 80.8%; **Figure 6A**). ET3 had the highest proportion of unique KOs (4.6%), followed by ET1 (2.6%), and ET2 (1.4%) indicating highest functional diversity for ET3 and lowest for the *E. coli* dominated ET2. Quantitatively, however, variation in

copies per million per KO were higher between ET3 and ET2 (27.2% of KOs significantly different) than between ET3 and ET1 (0.96% of KOs) indicating functional redundancy between the later despite pronounced taxonomic differences (**Figure 6B**). Significant differences were observed at the community level between the different enterotypes when comparing them using a PERMANOVA (**Figure 6C**). However, the difference was much more pronounced when comparing ET2 to ET1 and ET3 with R-squared values of 0.338 ($P = 0.001$) and 0.347 ($P = 0.001$), respectively, than between failure-associated ET1 and success-associated ET3 ($R^2 = 0.073$, $P = 0.009$). In **Figure 6D**, we show that the high-level metabolic potential of the community is sensibly similar between enterotypes and KOs involved in genetic information processing, metabolism, as well as signalling and cellular processes account for over 80% of the identified pathways for all enterotypes. We observed significant differences between enterotypes in specific pathways that were previously linked to positive STH infection status in the literature²²⁻²⁴. For instance, three pathways including glycine, serine and threonine metabolism, prokaryotic carbon fixation, as well arachidonic acid metabolism were significantly enriched in failure-associated ET1 and ET2 when compared to ET3 (**Figure 6E**, complete list of differentially abundant pathways in **Supplementary figure 7**).”

We discussed these results in lines 298-314:

“Finally, another potential mechanism of interaction could be driven by the metabolic potential at the community level. Currently, data about interactions between the gut microbiota and helminth colonization is rather scarce – the main source being observational cohorts comparing the metabolic make-up of infected and non-infected patients using high-throughput sequencing approaches. In this study, we highlighted community-specific differences in the abundance of metabolic pathways which – in turn – could potentially be linked to treatment outcome. While assessing the causal link will require subsequent studies, we show that the metabolic potentials of the three treatment-relevant enterotypes identified in this study present significant differences. This is especially true for failure-associated ET2 which presented profound differences with the two other enterotypes, ET1 and ET3. In addition, we found several specific pathways to be enriched in both failure-associated phenotypes - when compared to ET3 – that were previously associated with positive STH infection status. While this remains to be validated, we could hypothesize that certain

microbial compositions – and the associated metabolic traits – could provide a more or less permissive environment for the colonizing parasites and result in modulated efficacy of anthelmintic treatments.”

The corresponding methods are described from lines 420 to 428:

“Functional profiles were generated with Humann3⁴⁰ using default options in combination with the UniProt/Uniref 2019_01 database. The resulting gene abundance tables was converted to a KEGG orthologs (KOs) table using the humann_rename_table.py script. Upon renaming, tables were re-normalized to copies per Million using the humann_renorm_table.py script. Subsequently, KOs terms were regrouped into KEGG pathways using a custom KEGG pathway definition file. Finally, differential analysis of KOs was conducted using Deseq2⁴³ and group comparison was estimated using a custom fold-change script in combination with Kruskal-Wallis tests adjusted for multiple testing bias using the Benjamini-Hochberg procedure⁴⁴.”

3. The authors have broken down the data into purely categorical variables (cure vs non-cure). However, egg burden is a continuous variable and there is variation in responses that are more complex than cure vs non-cure. For example, “change in egg burden” i.e. there will be individuals where egg burden is reduced but not cure and others that are not reduced. Why would enterotypes not be associated with change in egg burden, but with cure vs non-cure?. Some discussion of this could be important.

We have added an additional analysis to measure associations between pre-treatment enterotype and egg reduction rate with a threshold for treatment success set to an ERR of 90%. We found that enterotype was also associated with ERR for both, *T. trichiura* and hookworm, when using ERR as a measure of treatment outcome (**Supplementary figure 4**). We have added the following text at line 177-182:

“We performed a similar analysis with a 90% egg reduction rate (ERR) as an estimate of treatment outcome. We found significant associations between enterotypes and ERR for both types of infection in the combination therapy arm similar to those found between enterotypes and cure rate (**Supplementary figure 4**).

We didn't find any associations between enterotype and ERR for patients receiving the monotherapy.”

4. It is a bit of a puzzle to me why the Hookworm cure rate is so low in this community. Most studies, as well as our own experience, typically describe much higher cure rates closer to 90-100%. The authors may want to provide some insight and discuss this in their discussion.

It is correct that the efficacy against hookworm based on cure rate was lower in Laos when compared to the other study settings of the multi-country trial administering albendazole-ivermectin. However, the egg reduction rate was >99%. The slightly lower cure rate might be due to the older age of the Laos participants. We would like to refer the reviewer to our recent publication for an in-depth discussion on the treatment efficacy ([https://doi.org/10.1016/S1473-3099\(21\)00421-7](https://doi.org/10.1016/S1473-3099(21)00421-7)).

5. It should be noted that ET3 is the most diverse of the enterotypes, the authors may want to comment on why the more diverse the bacterial community, the better they seem to achieve higher cure rates.

We thank the reviewer for this comment and have added a comparison of alpha diversity indices by enterotype to our manuscript (**Supplementary figure 3**). We observed a clear difference for two out of three indices – Shannon diversity and Berger-Parker dominance - when comparing ET1 and ET3 to ET2, but not when comparing ET1 to ET3.

We added the following text at lines 145-148:

“Comparison of alpha diversity showed significantly lower Shannon diversity and higher Berger-Parker dominance indices when comparing ET2 to ET1 and ET3 (**Supplementary figure 3**) but no differences were observed when comparing ET1 to ET3.”

MINOR COMMENTS

Results:

1. Line 120: “Among these, 86.2% (n=69) were also co-infected with hookworms, 40% 120 (n=32) with *A. lumbricoides*, and 8.7% (N=7) with *Opisthorchis viverrini*.”
Are there samples with only one infection?

Coinfections were highly prevalent in this cohort. We found only 1 patient with a *T. trichiura* monoinfection in the combination therapy arm and 3 in the monotherapy arm.

Methods:

1. Line 342-343: Author mentioned “The V3-V4 hypervariable region of the 342 16S rRNA gene was sequenced as described previously”, but in line 346 “The V4 region was amplified by cycling the reaction...” . Please edit.

This typo has been corrected.

Reviewer #3 (Remarks to the Author):

The research paper by Scheenberger et al approaches a relevant question in antihelminthic therapy and contributes with meaningful methodologic tools for the understanding of the variable and overall low cure rates of infections due to some species of STH. The establishment of links between treatment response and gut microbiome triggers very important hypothesis, and this report does it.

In order to make adequate interpretation of these results and identify follow-up questions, some clarifications are needed to fully support its conclusions.

Major comments:

The main analysis is based on 3 enterotypes; however, there is no clarity whether these is a pre-defined analysis or rather a post-hoc analysis based on the selected enterotypes when the results of cure rates were already available. The rules and criteria used for defining the 3 groups instead of other groups and/or a different number of enterotype groups needs to be clearly defined; also, whether the definition of these groups was done blinded of the treatment response results; along the same lines, the accuracy that is referred in line 134 should be further clarified.

We thank the reviewer for this important comment. We performed naïve enterotype classification before assessment of treatment outcome. We have added details on how the enterotypes were selected (see **comment 1 from reviewer 2**), which – in essence – was performed independently of the assessment of treatment response.

The accuracy at line 134 refers to the accuracy obtained when splitting the data into training and validation datasets and using a machine learning approach (random forest model) to predict classification of the samples in the validation subset. It is a post-hoc analysis to assess whether our classification system into the above mentioned enterotypes performs better than a random classification. In addition, the random forest model also enabled extracting the most relevant datapoints (= taxa) from the dataset that contributed to the enterotype classification (**Figure 2B**).

Although in line 204 it is stated that the findings remain after adjusting for infection intensity, a more robust adjustment on the same variable could be based on baseline egg count and its correlation to enterotype; perhaps a graph and analysis similar to what it is applied in Figure 3 of the supplement could be useful.

We thank the reviewer for this suggestion and have added an adjustment using the baseline egg count in **Table 2** (“Baseline EPG”) which remains consistent with the other bi-variate models.

We have also added a scatter plot to show the correlation between pre-/post-treatment EPG counts, by treatment arm (**Supplementary figure 6**). We described the results at line 214-216:

“Pre-/post-treatment EPG counts strongly correlated in the monotherapy arm, but did not correlate in the combination therapy arm, which is consistent with the observed treatment efficacies (**Supplementary figure 6**).”

Of note, we have also corrected the HR values showed in Table 2 and the corresponding in-text results as we previously reported the raw β coefficient output from R for *Trichuris trichiura* instead of the actual hazard ratios. The conclusions remain unchanged and the differences observed between enterotypes-associated cure rates are now increased (= increased hazard ratios) compared to what we initially reported.

The definition of “clearance” (and cure?) is stated as “two consecutive samples with no eggs (*Trichuris trichiura* or hookworm) detected and an average egg count between day 14 and 28 equal to zero” is confusing and hard to understand, since the finding of any amount of eggs equal or higher than 1 in any post treatment sample from day-14 on, should be considered as a treatment failure.

Please see our response to comment Nr 4 from reviewer 2. We have now added an analysis to measure association between enterotype and a threshold of treatment success based on a 90% egg reduction rate.

Minor comments

The abstract should include the sample size of each experimental group.

We have added the information in the abstract.

Ln75. The word “advertise” is probably inadequate for the intended meaning of that sentence.

We have replaced “advertised” with “described” to improve clarity.

Ln94-96. The sentence seems incorrect in its construction.

We have corrected the sentence in the revised manuscript.

The long discussion centered on the potential role of ivermectin as a substrate of macrolide-resistance as well as a promoter of drug resistance appears as too much speculative and although a thought-provoking idea, might be soften and supported by empirical data.

We have removed highly speculative parts of the discussion and added a section pertaining an interaction hypothesis based on the metabolic potential at the community level.

Reviewers' Comments:

Reviewer #2:

Remarks to the Author:

The authors have carefully addressed our comments from the initial submission. Most importantly, they have clarified 16S vs metagenomic classification of enterotypes and used the metagenomic data for some functional analysis. They have also addressed important concerns from other reviewers. I have no further comments.

Reviewer #3:

Remarks to the Author:

All major issues have been addressed and provided the technical aspects addressed in response to comments by Reviewer #2 to which are referred some of my major comments (and are beyond my area of expertise), the manuscript has the merits and originality to deserve publication.